# Improvement of Superconducting Joint Properties for GdBa_2_Cu_3_O_x_ Bulk Superconductors Joined with ErBa_2_Cu_3_O_x_ Superconductor Using Local Melt-Growth Method

**DOI:** 10.3390/ma17020484

**Published:** 2024-01-19

**Authors:** Kento Takemura, Kimiaki Sudo, Masaki Sakafuji, Kazuya Yokoyama, Tetsuo Oka, Naomichi Sakai

**Affiliations:** 1Regional Environment Systems, Shibaura Institute of Technology Toyosu Campus, 3-7-5 Toyosu, Tokyo 135-8548, Japan; mb20013@shibaura-it.ac.jp (K.S.); mb21501@shibaura-it.ac.jp (M.S.); okat@shibaura-it.ac.jp (T.O.); nsakai@shibaura-it.ac.jp (N.S.); 2Division of Electrical and Electronic Engineering, Ashikaga University Omae Campus, 268-1 Omae, Ashikaga 326-8558, Japan; yokoyama.kazuya@g.ashikaga.ac.jp

**Keywords:** superconducting joint, bulk superconductor, ErBCO, GdBCO, trapped magnetic field

## Abstract

The important factors in obtaining a high-quality superconducting joint were investigated for the superconducting joint of a GdBa_2_Cu_3_O_x_ (GdBCO) bulk superconductor with sintered ErBa_2_Cu_3_O_x_ (ErBCO) using the local melt-growth method. REBCO (RE: rare earth) bulk superconductors can be used as strong magnets by magnetizing them, but they require large bulk sizes for their application. Although the superconducting joint presents a viable solution, many possibilities for property improvement remain, such as property degradation, depending on the joining direction. By varying the joining thermal conditions and confirming the elemental distribution, magnetization properties near the joined part and the effects of these on the joining properties are clarified, and a method for fabricating high-performance joined samples is obtained. Microstructure segregation was rarely observed at the center of the joined part regardless of the joining direction, and the superconducting properties were negligible and small. The *J*_c_-*B* results are almost identical to those of the GdBCO matrix at a low field, confirming that the joined part minimally interferes with the superconducting current. Furthermore, by lowering the maximum temperature, shortening the holding time at the maximum temperature, and increasing the cooling rate, the region of mutual solid solution was reduced, and the *J*_c_-*B* under the self-magnetic field was enhanced. These results contribute to the development of the superconducting joining method, a critical aspect of larger bulk superconductors.

## 1. Introduction

A superconductor is a material that has zero electrical resistance when cooled to the superconducting transition temperature (*T*_c_: critical temperature). The REBCO bulk (melt-grown pseudo-crystal of the REBa_2_Cu_3_O_7−δ_ (RE123) superconductor with the RE_2_BaCuO_5_ (RE211) normal phase as the dispersion (RE: rare earth)) shows a *T*_c_ at around 90 K. By magnetizing the REBCO bulk material, a strong magnet can be fabricated. The YBCO bulk can achieve a trapped magnetic field of over 17 T at 29 K [1]. This performance is much higher than the surface flux of about 0.5 T for Nd-Fe-B permanent magnets. Therefore, many new applications have been proposed, such as magnetic bearings used for a non-contact rotating machine, a magnetic separation system, compact NMR, and superconducting bulk motors that generate strong torque [2,3,4,5,6,7,8,9,10,11]. Here, in order to increase the magnetic field trapped by the bulk superconductor, an increase in the critical current density (*J*_c_) and the size of the bulk is required. In addition, due to the short coherence length and high anisotropy of the RE123 superconductor, the *J*_c_ of the REBCO bulk is significantly reduced by the existence of weak links, such as dirty grain boundaries and a large misorientation angle, within the bulk [12]. To fabricate single-domain RE123 bulks without weak links, the top-seeded melt-growth method is generally employed. In this method, the REBCO raw material (RE123 + RE211) is melted above the peritectic decomposition temperature (*T*_p_) of RE123. Then, RE123 pseudo-crystal embedded with small RE211 particles is grown and oriented from the seed crystal placed on the surface during slow cooling. However, it is very difficult to fabricate a large and uniform high-quality superconducting bulk using the top-seeded melt-growth method [13]. This is because the crystal growth rate is slow, and microstructural changes, such as segregation and compositional change, tend to occur during crystal growth, which results in the degradation of the superconducting properties. The multi-seeded melt-growth method has been devised as one of the solutions [14,15,16,17,18]. In this method, multiple seed crystals are placed on a large precursor before the melt-growth process begins, and crystals are grown from each seed crystal to achieve larger dimensions. However, at the growth edges between crystals, a large number of second phases (RE211, etc.) is concentrated and superconducting properties are greatly reduced. Therefore, the bulk fabricated by the multi-seeded melt-growth method does not have a single peak in the magnetic field distribution when magnetized; however, it has multiple lower peaks that depend on the position of the seed crystals. Therefore, the superconducting joining method has been devised [19,20,21,22]. This is a technique to join multiple high-quality melt-grown bulks without the loss of superconducting properties. In this study, the local melt-growth method, one of the superconducting joining methods, was employed. The *T*_p_ of REBCO in air varies widely from about 880 to 1090 °C depending on the choice of RE elements (Nd, Gd, Y, Er, Yb, etc.). By inserting a RE’BCO (ex, RE’ = Er) thin plate, which has a lower *T*_p_ than REBCO (ex; RE = Gd), in between high-performance REBCO-oriented bulks, only the RE’BCO part below the *T*_p_ of REBCO experiences local melt growth; therefore, it is possible to obtain superconductively connected large REBCO bulks. It has been reported that the joining direction of the matrix may strongly affect the superconducting properties of the joined part in the local melt-growth method [22]. When the joining planes of the matrices were (110) planes and parallel to each other ((110)/(110) joint), a good superconducting joint was successfully obtained. However, when the joining planes were (100) planes and parallel to each other ((100)/(100) joint), the second phase and other phases were concentrated at the center of the joined part, and a degradation of superconducting properties was observed. The superconducting properties were compared using magneto-optical microscopy. Since magneto-optical microscopy can visualize the distribution of magnetic fluxes in terms of light and dark contrast, the quality of the joined part can be confirmed by the intensity of the external magnetic field applied. The (110)/(110) joint showed no magnetic flux penetration at an external magnetic field of 1000 Oe, while for the (100)/(100) joint, magnetic flux penetration was already visible in the joined part in an external magnetic field of 200 Oe. This clearly suggests that the joined parts of the (100)/(100) joint are poor quality. Previous reports mostly employed a slow cooling rate of 0.5 °C/h during melt growth and showed that long processing times sometimes caused problems. This slow cooling rate is the rate applied when fabricating melt-grown bulk with a diameter of approximately 20 mm or larger. It thus may not be optimal for melt-growing small regions such as superconducting joints. Therefore, investigating the effect of joining thermal conditions could improve the superconducting properties at the joined part.

In addition, upon controlling the superconducting property *J*_c_-*B* of the joined part, the joined part becomes the preferential field penetration path during pulse field magnetization, which is expected to increase the efficiency of the joined bulk superconductor and its application in various fields [23]. Thus, the local melt-growth method is useful for increasing the size of the REBCO bulk and has great potential as a new magnetization method and utility in new applications. Consequently, it is believed that by examining the joining thermal conditions during local melt growth in detail, there is significant potential for improving and controlling the properties of the joined part.

We attempted to investigate and identify the key factors that improve the superconducting joining properties for GdBCO bulk superconductors joined with an ErBCO superconductor using the local melt-growth method. In this study, we clarified the effects of the joining direction, maximum temperature (*T*_max_), holding time at maximum temperature (*t*_keep_), and cooling rate (*V*_cool_)—which are important conditions for superconducting the joint using the local melt-growth method—on the microstructure and superconducting joining properties. In order to make the important factors clear, microstructural observation and elemental distribution analysis using an electron probe microanalyzer (EPMA), as well as the characterization of superconducting properties (critical temperature and critical current density) using a superconducting quantum interference device (SQUID), were performed.

## 2. Materials and Methods

A GdBCO melt-grown bulk superconductor, a melt-grown pseudo-single crystal oriented in the [001] direction, was selected as the matrix used to fabricate the joined bulk superconductor. The dimensions were 60 mm O.D. and 40 mm height, and the composition was GdBa_2_Cu_3_O_7−δ_ (Gd123) and Gd_2_BaCuO_5_ (Gd211), each 5:2 in molar ratio, with 20 wt% Ag and 0.5 wt% Pt added. The ErBCO joining material was ErBa_2_Cu_3_O_7−δ_ (Er123) and Er_2_BaCuO_5_ (Er211) each 5:2 in molar ratio, with 10 wt% Ag_2_O and 1 wt% CeO_2_ added. Where Gd211 and Er211 act as a flux pinning center to improve *J*_c_, Ag is added to fill vacancies and cracks to improve the mechanical strength as fracture initiation points are reduced. The addition of Pt and CeO_2_ to the superconductor has the effect of increasing the frequency of RE211 nucleation during the melt reaction (RE123→RE211 + liq.) and reducing the diffusion rate of RE elements when RE211 is Ostwald-grown in the melt. Therefore, the grain growth of RE211 is suppressed [24,25,26]. The joining material was pressed in a mold and sintered in an electric furnace at 900 °C for 10 h in air. The *T*_p_ of both materials was measured via differential thermal analysis (DTA) (TG-DTA2020SA by BRUKER AXS, Yokohama, Japan) to be 1012 °C for the matrix and 945 °C for the joining material in the air. To prepare the joined samples, two 3 mm cubic pieces were cut from the GdBCO bulk superconductor with the (110) or (100) plane on the surface. The crystal orientation was confirmed by X-ray diffraction (XRD) (SmartLab by Rigaku Corporation, Tokyo, Japan). The surfaces to be joined were mirror-polished down to a 0.3 μm abrasive size using 3M lapping films, and ErBCO sintered material was polished to a 0.3 mm thickness and placed between the two GdBCO bulks. Figure 1 shows a schematic diagram of the joining thermal conditions. Figure 1a shows the setup of the precursor when set in the electric furnace. From bottom to top, the items are ordered as follows: alumina plate, MgO single crystal, matrix, joined part, and matrix. Figure 1b shows the heating program for the local melt-growth method. Here, the key conditions of the superconducting joint using the local melt-growth method are varied: the joining direction, *T*_max_, *t*_keep_, and *V*_cool_. *T*_max_ is set to exceed the *T*_p_ of the joined part, but not the *T*_p_ of the matrix, and *t*_keep_ is the holding time at *T*_max_. These parameters should be kept under conditions where only the joined part is sufficiently melted and the matrix is not degraded. The crystal growth of the joined part occurs in the slow cooling range of 940–920 °C. The quality of the joined part varies greatly with changes in *T*_max_, *t*_keep_, and *V*_cool_. Table 1 shows a list of samples with different joining directions and joining thermal conditions based on a matrix joining surface of (110), maximum temperature *T*_max_ = 980 °C, holding time at the maximum temperature *t*_keep_ = 3 h, and cooling rate *V*_cool_ = 1.67 °C/h. Here, the sample names “*T*_max_ 980”, “*t*_keep_ 3”, and “*V*_cool_ 1.67” are equivalent to “(110)/(110)”. The joined samples were annealed at 400 to 300 °C for 100 h under an oxygen atmosphere to make them superconducting.

First, the joined samples were molded using epoxy resin to obtain the (001) plane as the surface, and then the surface was mirror-polished down to a 0.3 μm abrasive size using 3M lapping films. Microstructural observation and compositional analysis of each region in the joined samples were performed by EPMA (EPMA-8050G by Shimadzu Corporation, Kyoto, Japan). Line analysis was performed to confirm the distribution of elements in the vicinity of the joined part and to determine the cause of degradation. Line analysis was performed at 400 µm perpendicularly to the joined part and 800 measurement points. Next, to confirm the overall superconducting properties of the fabricated joined sample, the magnetic field distribution was measured using the field cooling magnetization method (FCM). Here, a different sample was fabricated from the other measurements, with the same conditions as for the “(110)/(110)”, with a size of 10 × 10 × 3 mm^3^, and the [001] direction was the thinnest. A Nd-Fe-B magnet with a surface flux density of 0.48 T was placed directly above the joined sample, and the sample was cooled to 77 K by pouring liquid nitrogen and magnetizing it. After 15 min of magnetization, the Nd-Fe-B magnet was removed and the trapped magnetic field distribution was measured at a pitch of 1 mm over a 50 × 50 mm^2^ area by a Hall sensor installed on an X-Y scanning stage at 1 mm above the top surface of the joined sample. Finally, SQUID (MPMS-XL by Quantum Design Japan, Inc., Tokyo, Japan) was used to measure the superconducting properties (*T*_c_ and *J*_c_-*B*) of the joined samples. Samples were cut from the microstructurally observed joined samples to approximately 1.5 × 1.5 × 0.5 mm^3^, where the [001] direction was the thinnest such that the joined parts were centered and parallel to each other. *T*_c_ was determined from the magnetic susceptibility measurement as a function of temperature (*M*-*T* curve). *T*_c_ was defined as the temperature at which the magnetic susceptibility rapidly drops to zero as the sample is taken from 80 to 100 K in a magnetic field of 10 Oe. *J*_c_-*B* was obtained from the magnetization as a function of the applied magnetic field (*M*-*H* curve). Measurements were taken at 50 K in the magnetic field range from −5 to 5 T. The *J*_c_-*B* curves were calculated from *M*-*H* curves using an extended Bean model [27]. These measurements quantitatively evaluated the microstructural distribution and superconducting properties of the joined samples under different fabrication conditions.

## 3. Results

### 3.1. Microstructural Observation and Elemental Distribution of Joined Samples by EPMA

As a representative example, Figure 2 shows the results of microstructural observation and elemental distribution near the joined part by EPMA for the “(100)/(100).” The COMPO (reflected electron) image shows the difference in the atomic number of the element: black circles indicate voids, gray particles with a diameter of about 10 μm indicate Ag, and fine gray particles indicate the RE211 phases. Gd elements are located at the upper and lower parts, and Er elements are located at the center, the joined part. In the joined part, there was no significant segregation of RE211, and other undesirable phases, such as Ba-Cu-O, which would interfere with the superconducting current, were not observed. The microstructural observation of the “(110)/(110)” also shows a clean joining interface. Therefore, it was confirmed that, unlike in the previous report, it is possible to obtain a clean joining interface even in the case of the (110)/(110) joint in our joining condition. During RE123 crystal growth of a superconducting joint, crystals grow from both sides of the matrix. According to the “pushing–trapping theory” [28], small particles of RE211, etc., are pushed out into the liquid phase at the growth edge of RE123, depending on the growth conditions. Therefore, the small particles eventually concentrate at the center of the joined part. As shown in Figure 3 (modified from a previous report [22]), the crystal growth of the (100)/(100) joint at the joined part proceeds parallel to the matrix of the (100) plane. If the pushing effect occurs, the particles segregate in a line, and then the superconducting properties are reduced. In the (110)/(110) joint, the matrix surface is tilted at 45° from the (100) plane, so the crystal growth in the joined part proceeds in a triangular wave-like shape at the growth edges. The particles are trapped at the valleys of the triangular wave, the segregations are dispersed as point defects, and *J*_c_ at the joined part is not as degraded as in the previous report [22]. Here, the pushing effect tends to occur at a slower growth rate. The cooling rate in the previous report was slow (0.5 K/h), while the cooling rate in this study was fast (1.67 K/h). This allowed us to obtain a clean joining interface, even for the (100)/(100) joint. The faster cooling rate increased the crystal growth rate, facilitating the trapping of RE211 particles in the RE123 crystal based on the pushing-trapping theory. Therefore, a good superconducting joint without significant segregation has been successfully obtained regardless of the joining direction.

As per Figure 2, the overlap of Gd and Er elements was observed at the interface between the matrix and the joined part. This indicates the existence of a region of the mutual solid solution (MSS). This is thought to be caused by the back-melting of the GdBCO matrix during local melt growth at high temperatures, and the MSS can degrade superconducting properties.

Figure 4a,b shows a compositional image and line analysis of the Gd and Er elements in the (001) plane of “(100)/(100),” respectively. In Figure 4a, the region used for line analysis is indicated on the COMPO image. In Figure 4b, the green line shows the intensity of Gd, and the blue line shows the intensity of Er. The multiple sharp, high peaks indicate the RE211 particles embedded in the RE123 phase. The MSS regions with the intensity gradient of Gd and Er are observed at the interface between the GdBCO matrix part and the ErBCO joined part. To define the MSS length, the change in the Gd intensity was used. First, the Gd intensity data were rounded using the moving average of 20, as shown by the red line in Figure 4b. MSS was defined as the range where the average of the matrix part intensity was reduced to 90% and 30%. Here, almost constant amounts of Gd elements are observed in the joined part. It is considered that Gd goes into Er-Ba-Cu-O and then forms a solid (Er_1−x_, Gd_x_)-Ba-Cu-O-type solution. In addition, it is also possible that the Er-(Ba_1−y_, Gd_y_)-Cu-O-type solid solution was formed by melt growth in the air. This solid solution has a low *T*_c_, which leads to a decrease in the superconducting properties of the joined part [29]. The MSS length indicates the amount of back-melting in the matrix, and the formation is solid solutions; therefore, longer MSS lengths possibly correlate with decreased superconducting properties.

Figure 5a–c show the average MSS length for the samples joined in various thermal conditions. Here, the line analyses were repeated three times for each sample, and the average values were used.

Figure 5a shows that the MSS length increased with higher *T*_max_ and took on a saturating behavior. At *T*_max_ lower than 980 °C, the melted joining material does not have higher heat, has less back-melting, and has a shorter MSS length. Figure 5b shows that the MSS length increased rapidly as *t*_keep_ gets longer. The length of *t*_keep_ directly affects the time matrix in contact with the melted joining material, so the change in the MSS length is significant. Figure 5c shows that the MSS length decreased with faster *V*_cool_. The slow cooling range under joining thermal conditions is 940–920 °C, which is lower than the temperature range discussed in the change of *T*_max_ and *t*_keep_, but the MSS length was changed. Therefore, back-melting is considered to be occurring even during slow cooling, and it is promoted when *V*_cool_ is slow. This suggests the possibility of improving the quality of superconducting joints by lowering *T*_max_, shortening *t*_keep_, and making *V*_cool_ faster. However, there is a range of applicability that exists for *T*_max_ and *t*_keep_.

### 3.2. Superconductivity Properties of Joined Samples

Figure 6 shows the results of trapped magnetic field distribution measurements for the slightly large sample of 10 × 10 × 3 mm^3^ fabricated with the same joining condition as the sample “(110)/(110)”. The horizontal dotted red line at the center indicates the position just above the joined part. There is only one peak of the magnetic field and no significant decrease in the magnetic field at the dotted red line. This means the superconducting current flows across the joined part of the whole sample. In this experiment, the measurement pitch was 1 mm, which is larger than the thickness of the joined part (0.3 mm). However, in the case of poor joining, the superconducting current flowing in the joined part is suppressed, and two clear peaks are observed. This qualitatively indicates that the superconducting joint was successfully obtained in this study.

Figure 7 shows the temperature dependence of magnetic susceptibility measured by SQUID for GdBCO and ErBCO bulk, the “(110)/(110)” joined sample, and the “(100)/(100)” joined sample. The SQUID samples of GdBCO and ErBCO were cut from samples fabricated under the same joining thermal conditions with the sample “(110)/(110)” with a thicker joined part, as shown in the schematic diagram in Figure 7. *T*_c_ of GdBCO and ErBCO bulks are 94.0 K and 90.8 K, respectively. *T*_c_ of “(110)/(110)” and “(100)/(100)” joined samples shows little difference and a two-step transition at around 94 K and 91 K. The higher *T*_c_ around 94 K indicates the property from the GdBCO matrix part, and the lower *T*_c_ around 91 K indicates the property from the ErBCO joined part. The height of the shoulder depends on the volume of the joined samples, indicating that both joined samples contained about 25% of the joined part. Here, the lower *T*_c_ for the joined samples is shifted to a 0.5–1 K higher temperature than ErBCO’s, but the transition curves below *T*_c_ are not so different between the joined samples and ErBCO. Generally, *T*_c_ increases when increasing the size of RE elements (Gd is larger than Er), and superconducting transition behavior below *T*_c_ indicates the homogeneity of the sample [30]. Therefore, when the Er-(Ba_1−y_, Gd_y_)-Cu-O-type solid solution formed, *T*_c_ was decreased and the transition became broader, while in the case of the (Er_1−x_,Gd_x_)-Ba-Cu-O-type solid solution, *T*_c_ changed with the composition of Gd/Er. However, the transition behavior did not change much. Therefore, in “(110)/(110)” and “(100)/(100)” joined samples, as explained in the previous section, Gd diffuses into the joined part and forms a (Er_1−x_, Gd_x_)-Ba-Cu-O-type solid solution, resulting in an improved *T*_c_ with a sharp superconducting transition.

Figure 8a–c shows the temperature dependence of magnetic susceptibility measured by SQUID for the joined samples fabricated under various joining conditions. In Figure 8a, there is no significant difference in *T*_c_ for the samples “*T*_max_ 960“ and “*T*_max_ 980”, with these samples showing a sharp two-step transition with the higher *T*_c_ around 94 K and the lower *T*_c_ around 91 K. However, the sample “*T*_max_ 1000” shows different behavior: it exhibits lower *T*_c_ around 93 K with a broad superconducting transition. This may be due to the formation of the Er-(Ba_1−y_, Gd_y_)-Cu-O-type solid solution, as explained in the previous section. In Figure 8b, there is no significant difference in *T*_c_ for the samples “*t*_keep_ 1” and “*t*_keep_ 3”; these samples showed sharp two-step transitions, where the higher *T*_c_ and the lower *T*_c_ show similar values at around 94 K and 91 K. However, in the case of the sample “*t*_keep_ 5”, although it showed higher *T*_c_ at around 94 K, the superconducting transition was slightly broad, and the lower *T*_c_ decreased to around 90 K. This is also considered to be due to the occurrence of low *T*_c_ phases of the Er-(Ba_1−y_, Gd_y_)-Cu-O-type solid solution, the same as the sample “*T*_max_ 1000”. In Figure 8c, the effect of changing *V*_cool_ on *T*_c_ is small; all samples show sharp two-step transitions, and the higher *T*_c_ is almost the same, at about 94 K. The lower *T*_c_ also shows a similar value of about 91 K, but when processed with a faster cooling rate like “*V*_cool_ 20”, the lower *T*_c_ slightly shifts to a lower temperature. This may be caused by the formation of (Er_1−x_, Gd_x_)-Ba-Cu-O-type solid solution with smaller Gd substitution. The above results suggest that *T*_max_ thst is too high or *t*_keep_ that is too long promotes the formation of the Er-(Ba_1−y_, Gd_y_)-Cu-O-type solid solution and results in a superconducting joint with a low *T*_c_ phase.

Figure 9 shows the *J*_c_-*B* curves for the GdBCO bulk, ErBCO bulk, “(110)/(110)” joined sample, and “(100)/(100)” joined sample. In ErBCO, *J*_c_ rapidly decreased up to 0.5 T, and then *J*_c_ gradually decreased above 0.5 T. This is likely due to the surface flux pinning of the finely dispersed RE211 particles embedded in the RE123 superconducting phase. Meanwhile, in GdBCO, although *J*_c_ rapidly decreased to 0.5 T, however above 0.5 T, *J*_c_ gradually increased. This phenomenon is called the peak effect and is due to the formation of finely dispersed low *T*_c_ phases, like a small Er-(Ba_1−y_, Gd_y_)-Cu-O-type solid solution area embedded in a high-*T*_c_ RE123 phase [30]. This is because the ion size of Gd is slightly closer to that of a Ba ion, and Gd substitutes for the Ba site in RE123 crystals. This area is a superconductor, but because of its low *T*_c_ phase, it changes to a normal conductor at high magnetic fields and functions as a pinning center. This is generally known as a magnetic field-induced pinning center. The difference between “(110)/(110)” and “(100)/(100)” is almost negligible, with a maximum of ±15%. In the low field region up to 0.5 T, the *J*_c_ values are almost the same as the GdBCO bulk, which means that the *J*_c_ of the joined samples below 0.5 T is restricted by the *J*_c_ of the matrix GdBCO. However, *J*_c_-*B* curves of “(110)/(110)” and “(100)/(100)” are lower than that of GdBCO above 0.5 T. This is thought to be because the *J*_c_ through the entire sample is tied to the weakest properties within the sample. Therefore, in the case of the joined sample, *J*_c_ is equivalent to GdBCO at the low field below 0.5 T. However, in the high field above 0.5 T, the *J*_c_ of the joined part of “(110)/(110)” and “(100)/(100)” is lower than the *J*_c_ of GdBCO, resulting in this *J*_c_-*B* curve. Although this result is generally favorable, there is concern about the degradation of the joined part depending on the joining thermal conditions. Therefore, it is important to determine the effect and clarification of the joining thermal condition modification on *J*_c_-*B* improvement in the joined samples.

Figure 10a–c shows the *J*_c_-*B* curves of the samples joined under various melt-growth conditions. When *T*_max_ is 1000 °C, *J*_c_-*B* is extremely reduced. This may be due to the formation of an Er-(Ba_1-y_, Gd_y_)-Cu-O-type solid solution with low *T*_c_, as discussed in the *T*_c_ measurement, which also reduced *J*_c_-*B*. Therefore, *T*_max_ should be better to set at lower temperatures than 980 °C. Figure 10b shows that longer *t*_keep_ results in an overall decrease in *J*_c_-*B*. In particular, *J*_c_-*B* decreases significantly at “*t*_keep_ 5”. This decrease is also due to the formation of an Er-(Ba_1-y_, Gd_y_)-Cu-O-type solid solution. This suggests that shorter *t*_keep_ may improve *J*_c_-*B*. Figure 10c shows that a faster *V*_cool_ significantly increases *J*_c_ values in the low field region below 0.5 T. Generally, the *J*_c_-*B* property in the REBCO bulk superconductor at low magnetic fields below 0.5 T is proportional to *V*_f_/*d*, where *V*_f_ is the volume fraction of RE211 particles embedded in RE123 and *d* is the diameter of RE211 particles. Here, the size of RE211 particles tends to increase during the melting process due to the “Ostwald ripening”, and especially at higher temperatures and longer melting times, the grain growth of RE211 particles in the Ba-Cu-O melt is greatly enhanced [31]. Therefore, it is expected that the size of RE211 particles is similarly increased at higher temperatures and longer melting times during the joining process when using the local melt-growth method. The fast cooling rate “*V*_cool_ 20” shows a 55% increase from the *J*_c_ of “*V*_cool_ 1.67” under the self-magnetic field. On the other hand, “*V*_cool_ 0.5” shows the same trend in *J*_c_-*B* as “*V*_cool_ 1.67” at low magnetic fields below 0.5 T; however, a large *J*_c_ value with a peak effect is obtained at high magnetic fields above 0.5 T. The difference in growth temperature between the Gd-Ba-Cu-O high *T*_c_ phase and the Gd-(Ba_1−y_, Gd_y_)-Cu-O low *T*_c_ phase was very small, and a slightly lower *T*_c_ phase was uniformly obtained when the GdBCO sample was melt-grown in air. However, when the GdBCO sample was melt-grown under a very slow cooling rate, Gd/Ba substitution was basically restricted and obtained a high *T*_c_ phase. Here, the high *T*_c_ GdBCO matrix sometimes contained a nanometer-sized low *T*_c_ area [26]. Therefore, unlike other samples, “*V*_cool_ 0.5” is thought to exhibit a peak-effect *J*_c_-*B* at high magnetic fields above 0.5 T.

The results of *J*_c_-*B* measurements with different joining thermal conditions correlate with the trend in MSS length, suggesting that lowering *T*_max_, shortening *t*_keep_, and making *V*_cool_ faster may improve the quality of the superconducting joint. However, careful adjustment is needed in setting the joining thermal conditions. This is because there is a temperature range in which only the joined part is melted. In this study, the *T*_p_ of the joined part is 945 °C and the *T*_p_ of the matrix is 1012 °C, so *T*_max_ can only be adjusted between them. Even within the range, melting at high temperatures and for long periods of time will cause property degradation due to compositional shifts caused by liquid loss and grain growth of the RE211 phase. In addition, if the combination of *T*_max_ and *t*_keep_ is not sufficiently adjusted, not enough heat will be applied to the center of the joined part, and the joined part may not melt. Especially when a large bulk is joined, O_2_ gas remains from the reaction (RE123→RE211 + Liq. + O_2_) during melting, which increases *T*_p_, and thus the process must be carried out at a higher temperature. If *V*_cool_ is too slow, liquid loss and coarsening of the RE211 phase will cause *J*_c_ reduction. Additionally, if *V*_cool_ is too fast, the growth rate may not keep up, resulting in non-uniform nucleation and polycrystalline. At the same time, however, since the area of melt growth in the local melt-growth method is only the joined part and not the entire bulk as in the melt-grown bulk, a low temperature, short duration, and fast cooling may be sufficient for melting.

## 4. Conclusions

We investigated and clarified the important factors used to improve the superconducting joining properties for a GdBCO bulk superconductor joined with an ErBCO superconductor using the local melt-growth method. We first clarified the effects of the joining direction, *T*_max_, *t*_keep_, and *V*_cool_—which are important conditions needed in the superconducting joint when using the local melt-growth method—on the microstructure and superconducting joining properties. Controlling the following three factors by changing the above four joining conditions can improve the superconducting joining properties: (1) the elimination of segregation of undesirable phases RE211 and Ba-Cu-O; (2) elimination and control of the solid-solution formation; and (3) size control of the pinning center, RE211, etc. More specifically, the following results were found.

A good superconducting joint was successfully obtained in the (100)/(100) joint, which was almost the same as that in the (110)/(110) joint. In previous reports, a good superconducting joint could not be obtained in the (100)/(100) joint due to degradation caused by impurity segregation. Still, by employing a faster cooling rate, the segregation of impurities was suppressed.The formation of the solid solution in the Gd/Er/Ba site sometimes resulted in the degradation of *J*_c_. The MSS (mutual solid solution) length indicates the amount of back-melting in the matrix, and the formation is solid solutions; therefore, longer MSS lengths possibly correlate with decreased superconducting properties. EPMA line analysis revealed that the MSS length in the joined part varied depending on the joining thermal conditions. It was possible to decrease the MSS length in the joined part by lowering *T*_max_, shortening *t*_keep_, and making *V*_cool_ faster.The results of *T*_c_ and *J*_c_-*B* characteristics showed little difference depending on the joining direction, especially in the low field region of *J*_c_-*B*, which was equivalent to that of matrix GdBCO. In addition, the superconducting properties of the “(110)/(110)” were very good in the results of the trapped magnetic field distribution obtained by FCM, and it was confirmed that the joined part does not become defective when used as a bulk.*T*_c_ and *J*_c_-*B* results with different joining thermal conditions show that the local melt-growth method can be significantly degraded by the joining thermal conditions. Additionally, a good superconducting joint can be obtained by lowering *T*_max_, shortening *t*_keep_ and making *V*_cool_ faster.

The results of this study will contribute to the development of the superconducting joining method, which is an important key to the technology of increasing the size of bulk superconductors.

## Figures and Tables

**Figure 1 materials-17-00484-f001:**
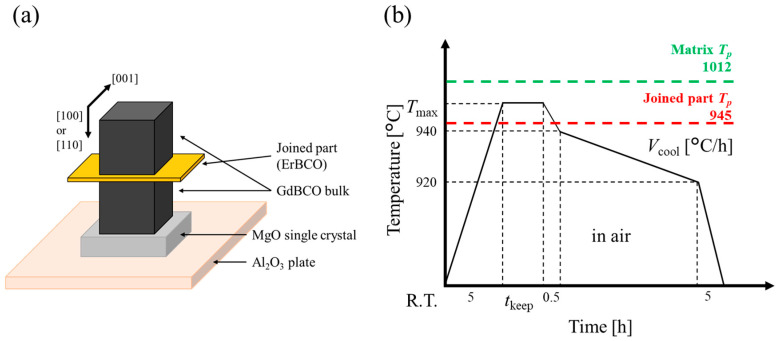
(**a**) Schematic diagram of precursors used in preparing joined sample. (**b**) Heat program used for local melt-growth method. *T*_max_, *t*_keep_, and *V*_cool_ were changed. *T*_max_: maximum temperature; *t*_keep_: holding time at the maximum temperature; *V*_cool_: cooling rate.

**Figure 2 materials-17-00484-f002:**
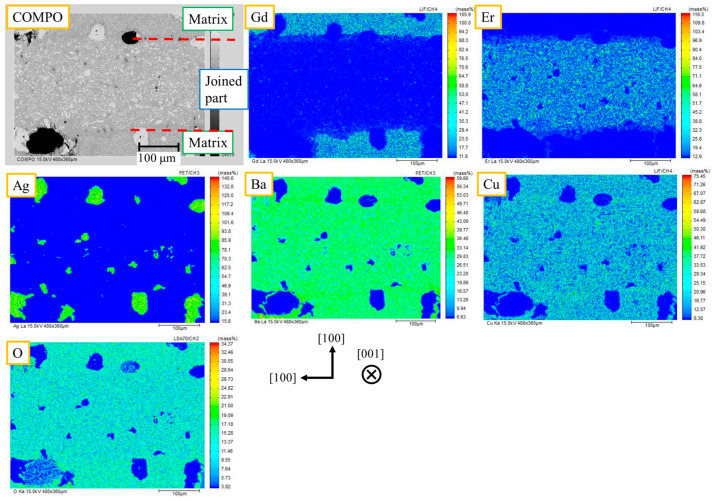
COMPO image and elemental distribution images of sample “(100)/(100)”. There is no significant concentration of RE211 or Ba-Cu-O phases at the center of the joined part.

**Figure 3 materials-17-00484-f003:**
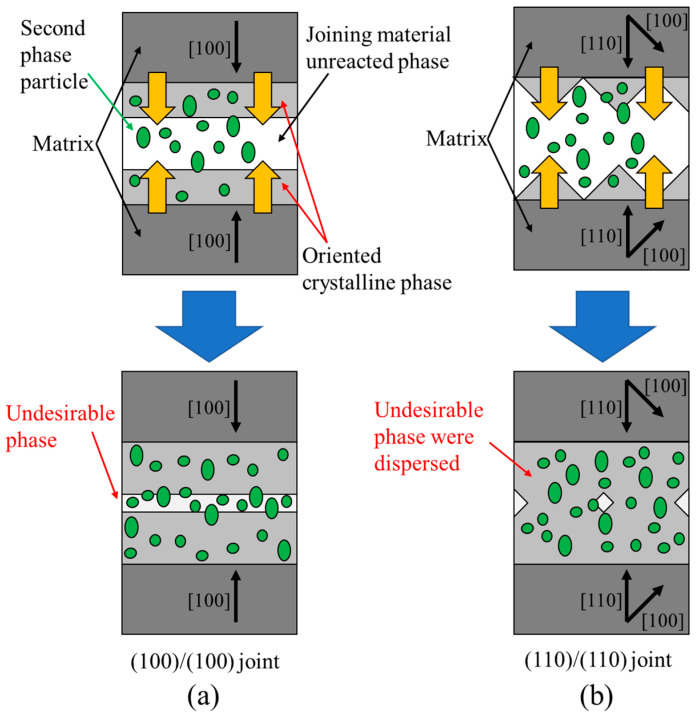
Schematic diagram of crystal growth of (**a**) (100)/(100) and (**b**) (110)/(110) joint. In the (100)/(100) joint, crystal growth at the joined part proceeds parallel to the matrix, and segregation is concentrated at the center of the joined part. In the (110)/(110) joint, crystal growth proceeds in a triangular wave-like shape, and segregation is dispersed (modified from [22]).

**Figure 4 materials-17-00484-f004:**
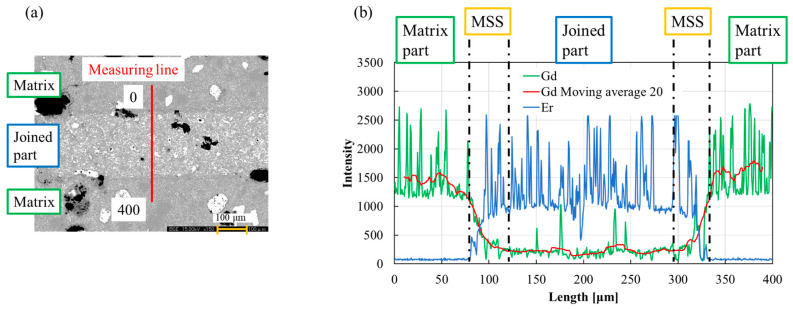
The results of EPMA line analysis of the “(100)/(100)”. (**a**) COMPO image showing the position of line analysis range of 400 μm. (**b**) Intensity of Gd and Er elements in line analysis. The red line is the moving average 20 of the Gd elements, and the MSS (mutual solid solution) boundaries are 90% and 30% from the Gd average intensity in the matrix part.

**Figure 5 materials-17-00484-f005:**
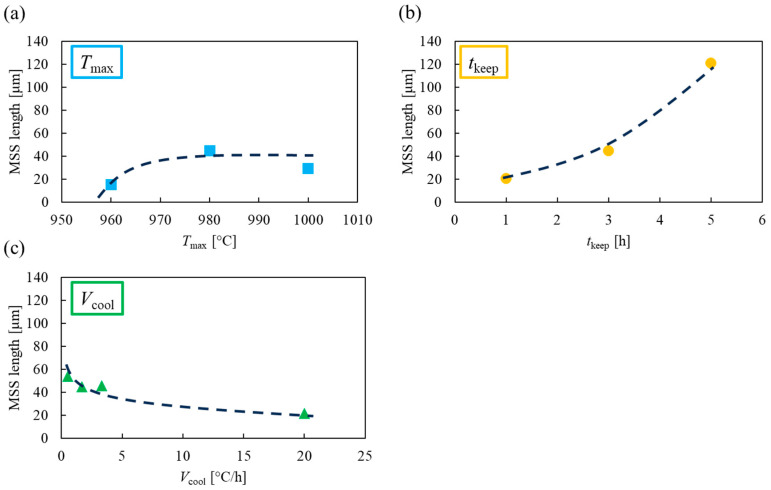
MSS (mutual solid solution) length vs. various joining thermal conditions. (**a**) *T*_max_: maximum temperature. (**b**) *t*_keep_: holding time at the maximum temperature. (**c**) *V*_cool_: cooling rate. Here, *T*_max_ is based on 980 °C, *t*_keep_ is 3 h, and *V*_cool_ is 1.67 °C/h. For example, for “*T*_max_ 1000”, only *T*_max_ is changed to 1000 °C.

**Figure 6 materials-17-00484-f006:**
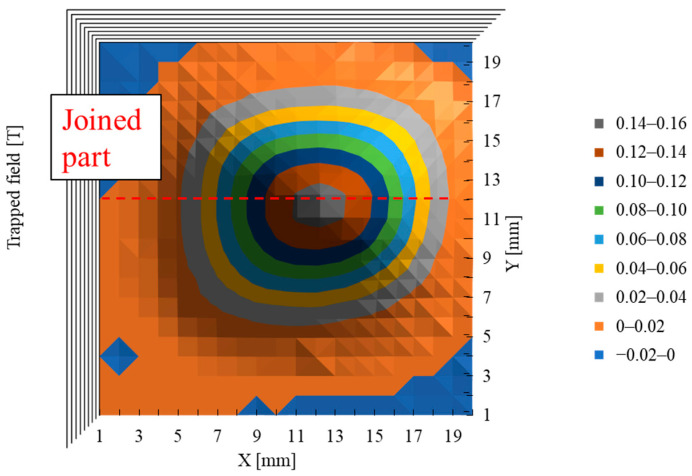
Trapped magnetic field distribution of the 10 × 10 × 3 mm^3^ sample processed by the same condition as the “(110)/(110).” The sample was magnetized by Nd-Fe-B magnet of 0.5 T parallel to the [001] direction at 77 K and measured 1 mm above the sample. The dotted red line indicates the position of the joined part. There is only one peak and no decrease at the joined part.

**Figure 7 materials-17-00484-f007:**
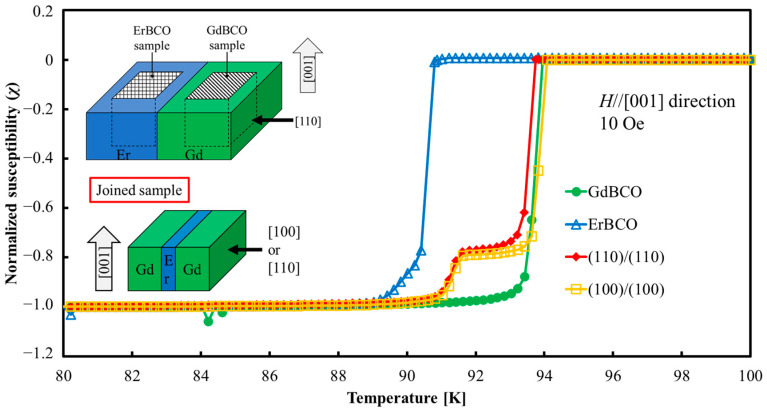
Temperature dependence of magnetic susceptibility measured by SQUID for GdBCO bulk, ErBCO bulk and for “(110)/(110)” joined sample and “(100)/(100)” joined sample. Inserted schematic diagram shows the cut-out positions of GdBCO, ErBCO, and joined samples. “(110)/(110)” and “(100)/(100)” show sharp two-step transitions around 94 K and 91 K, which indicate the properties of GdBCO matrix and ErBCO joined parts, respectively.

**Figure 8 materials-17-00484-f008:**
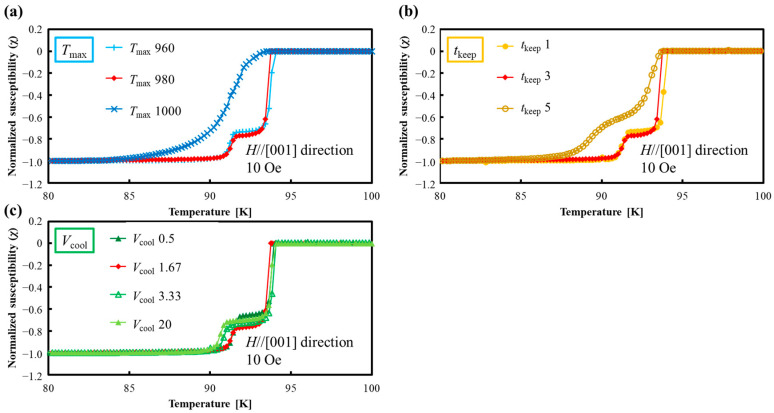
Temperature dependence of magnetic susceptibility of the samples joined under various melt-growth conditions. (**a**) *T*_max_: maximum temperature. (**b**) *t*_keep_: holding time at the maximum temperature. (**c**) *V*_cool_: cooling rate. Broad transition shows the formation of Er-(Ba_1−y_, Gd_y_)-Cu-O-type solid solution. The sample names “*T*_max_ 980”, “*t*_keep_ 3”, and “*V*_cool_ 1.67” are equivalent to “(110)/(110)”. Here, *T*_max_ is based on 980 °C, *t*_keep_ is 3 h, and *V*_cool_ is 1.67 °C/h. For example, for “*T*_max_ 1000”, only *T*_max_ is changed to 1000 °C.

**Figure 9 materials-17-00484-f009:**
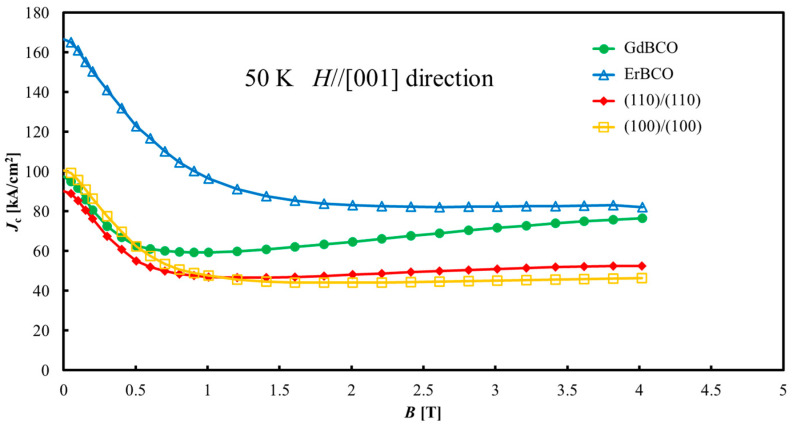
*J*_c_-*B* curves for GdBCO bulk, ErBCO bulk, “(110)/(110)” joined sample, and “(100)/(100)” joined sample at 50 K. The difference between “(110)/(110)” and “(100)/(100)” is small, and in the low field region up to 0.5 T, the *J*_c_ values are not inferior to those of GdBCO, which is a matrix.

**Figure 10 materials-17-00484-f010:**
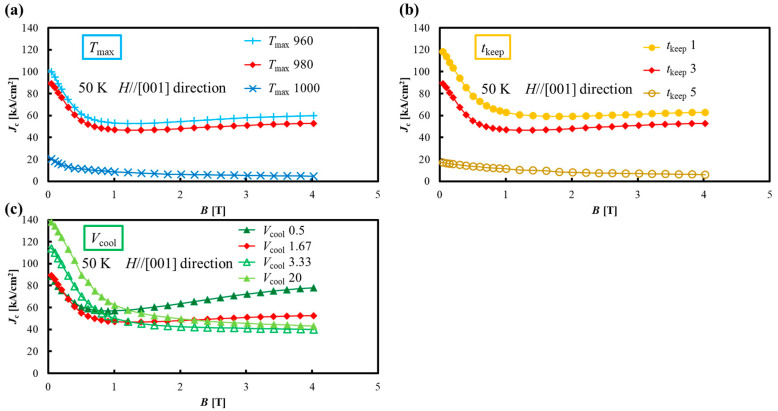
*J*_c_-*B* curves of the samples joined under various melt-growth conditions. (**a**) *T*_max_: maximum temperature. (**b**) *t*_keep_: holding time at the maximum temperature. (**c**) *V*_cool_: cooling rate. The sample names “*T*_max_ 980”, “*t*_keep_ 3”, and “*V*_cool_ 1.67” are equivalent to “(110)/(110)”.

**Table 1 materials-17-00484-t001:** List of the fabrication conditions used for superconducting joining samples with different joining thermal conditions. Bold type indicates items that differ from “(110)/(110)”.

Sample Name	(110)/(110)	(100)/(100)	*T*_max_ 960	*T*_max_ 1000	*t*_keep_ 1	*t*_keep_ 5	*V*_cool_ 0.5	*V*_cool_ 3.33	*V*_cool_ 20
Direction	[110]	**[100]**	[110]	[110]	[110]	[110]	[110]	[110]	[110]
*T*_max_ [°C]	980	980	**960**	**1000**	980	980	980	980	980
*t*_keep_ [h]	3	3	3	3	**1**	**5**	3	3	3
*V*_cool_ [°C/h]	1.67	1.67	1.67	1.67	1.67	1.67	**0.5**	**3.33**	**20**

## Data Availability

Data are contained within the article.

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
