# Peer review of "Improvement of Superconducting Joint Properties for GdBa2Cu3Ox Bulk Superconductors Joined with ErBa2Cu3Ox Superconductor Using Local Melt-Growth Method"

_materials, 2024, doi:10.3390/ma17020484_

Round 1
Reviewer 1 Report
Comments and Suggestions for Authors
K. Takemura et al. present work to improve superconducting joint properties of GdBa2Cu3O6+x with ErBa2Cu3O6+x superconductors using local melt-growth method. This study presents a significant improvement over previous attempts of joining REBa2Cu3O6+x samples with aims of developing methods for joining superconductors, which is important for producing large size bulk superconductors for various applications.
The scientific work is systematic and the results are clear. However, I recommend some improvements to some of the writings and descriptions of the figures to further clarify the conclusions before publication. I recommend publication of the work after these improvements.
YBCO is well known in the field. However, BCO are all elements and maybe confusing to some readers at first. It may be worthwhile to keep the title as GdBa2Cu3Ox and ErBa2Cu3Ox, then define in the beginning of the text that BCO refers to Ba2Cu3O6 to avoid confusion.
Comments on Figures:
Fig. 2 (b) Highlights of text for T_max, t_keep, and V_cool seems strange
For Fig. 5, I recommend clearly mentioning the t_keep and V_cool for (a), the T_max and V_cool for (b), and T_max and t_keep for (c) in the figure caption.
Table 1 borders extend outside the page.

Comments on the Quality of English LanguageI highlight some sentences that need improvement in the writing in the attached PDF.
Author Response
Thank you very much for your peer review.
I have corrected all the points you highlighted in PDF.
Additionally, I have improved the text to clearly explain the experimental method to the results.
Corrected GdBCO and other expressions including the title.
Removed highlights from Fig. 1. (b).
The unaltered condition was entered in Fig. 5 in the caption.
Table 1 is contained within the page.

Reviewer 2 Report
Comments and Suggestions for Authors This article investigated and made clear the important factors to improve the superconduct- ing joining properties for GdBCO bulk superconductor joined with ErBCO superconductor using the local melt-growth method. By clarifying the effects of the joining direction, Tmax, tkeep, and Vcool, which are important conditions for superconducting joint by the local melt-growth method, on the microstructure and superconducting joining properties, it was found that to control the following three factors by changing above four joining conditions are important to improve the superconducting joining properties. (1) elimination of segregation of undesirable phases RE211 and Ba-Cu-O, (2) elimination and control of the solid-solution formation, (3) size control of pining center, RE211 etc. These results will contribute to the development of the superconducting joining method, a critical aspect of larger size of bulk superconductors. However, before making the announcement, there are a few things to consider:1、Why does the addition of Ag increase mechanical strength, and the addition of Pt and CeO2 suppress the grain growth of the RE211 phase? Please explain in detail.
2、Why MSS may lead to the degradation of superconductivity. Can you find some theoretical evidence?
3、Figure 9 shows the decline in Jc. It is suggested that this may be caused by the surface flux pegging of dispersed RE211 particles embedded in the superconducting phase of RE123. Can you find some theoretical evidence?
4、The authors are suggested to compare some recent related works, (i.e. Applied Materials & Interfaces, 9(5), 2101933, (2022); Applied Materials Today. 28, 101546, (2022);
Comments on the Quality of English LanguageMinor editing of English language required.
Author Response
Thank you very much for your peer review.
I revised the entire text and made corrections.
I have indicated the answers to your questions and reflected them in my manuscript.
- Why does the addition of Ag increase mechanical strength, and the addition of Pt and CeO2 suppress the grain growth of the RE211 phase? Please explain in detail.
Our answer
I made an addition near line 112.
Ag is added to fill vacancies and cracks to improve mechanical strength as fracture initiation points are reduced.
The addition of Pt and CeO2 to the superconductor has the effect of increasing the frequency of RE211 nucleation during the melt reaction (RE123→RE211 + liq.) and reducing the diffusion rate of RE elements when RE211 is Ostwald grown in the melt.
Therefore, it has been reported that the grain growth of RE211 is suppressed.
- Why MSS may lead to the degradation of superconductivity. Can you find some theoretical evidence?
Our answer
I made an addition near line 234.
MSS is caused by the back melting of the GdBCO matrix during melt growth, which may result in the Er-(Ba1-y, Gdy)-Cu-O type solid solution described in the manuscript, causing degradation of superconducting properties.
The text has been improved to clearly explain this information.
- Figure 9 shows the decline in Jc. It is suggested that this may be caused by the surface flux pegging of dispersed RE211 particles embedded in the superconducting phase of RE123. Can you find some theoretical evidence?
Our answer
Around line 381, the literature [31] is cited for explanation.
The Jc-B property of REBCO bulk superconductors at low magnetic fields is proportional to Vf/d where Vf is the volume fraction of RE211 particles embedded in RE123 and d is the diameter of RE211 particles.
The size of RE211 particles may increase during the melting process due to the "Ostwald-Ripening effect," which greatly enhances the grain growth of RE211 particles.
- The authors are suggested to compare some recent related works, (i.e. Applied Materials & Interfaces, 9(5), 2101933, (2022); Applied Materials Today. 28, 101546, (2022);
Our answer
I have checked the papers you listed. However, these are not related to this paper. Perhaps you are indicating these papers. Please check them.
Applied Materials & Interfaces, 9(5), 2101933, (2022)
-> “MOF-Derived Porous Dodecahedron rGO-Co3O4 for Robust Pulse Generation” Xiaohui Li, Mingqi An, Gang Li, Yueheng Han, Penglai Guo, Enci Chen, Jiyi Hu, Zhuoying Song, Hongbing Lu, Jiangbo Lu.
Applied Materials Today. 28, 101546, (2022)
-> “Optical-intensity modulators with PbTe thermoelectric nanopowders for ultrafast photonics” Xiaohui Li, Wenshuai Xu, Yamin Wang, Xiaoli Zhang, Zhanqiang Hui, Han Zhang, S. Wageh, Omar A. Al-Hartomy, Abdullah G. Al-Sehemi.
Additionally, no superconducting joining bulk studies have been reported in recent years because it was believed that there was no potential for improvement beyond the previous studies.

Reviewer 3 Report
Comments and Suggestions for Authors
The manuscript is interesting, but I have some comments:
1) A flow chart of the measurements to characterize the sample is not very clear? What are the crucial experiments? A sort of flow chart should be discussed in the methodology
2) The authors do not show a current/temperature measurement? Is it worthy?
3) Could the authors mentioned something on critical parameters of the samples?
4) The sentence in the coclusion "A good superconducting joint was successfully obtained in (100)/(100) joint" is a bit puzzling and not quantitative. In general, the authors could try yo reword the conclusion section.
Author Response
Thank you very much for your peer review.
I have indicated the answers to your questions and reflected them in my manuscript.
- A flow chart of the measurements to characterize the sample is not very clear? What are the crucial experiments? A sort of flow chart should be discussed in the methodology.
Our answer
The explanation is given after line 149.
A clear explanation of the measurement procedure and how to prepare the measurement samples was added. The reason for conducting the measurements in the order of microstructure and composition distribution, trapped field distribution measurement, Tc measurement, and Jc-B measurement is that the first step is to observe the microstructure before machining the sample into a SQUID sample, so that a wide area can be observed. Next, trapped field distribution measurements were performed to qualitatively confirm that the joined part exhibits good superconducting properties. Finally, measurements were performed to quantitatively compare superconducting properties.
- The authors do not show a current/temperature measurement? Is it worthy?
Our answer
Since it is important to show that the superconducting current flows through the joined part in this experiment, trapped field distribution and Jc-B characteristics were evaluated.
In addition, the Jc-B measurement, a magnetization measurement method that can easily determine the degree of degradation of a superconducting joint, was adopted this time.
In the future, when more precise critical current values are required, the current measurement method may also become important.
- Could the authors mentioned something on critical parameters of the samples?
Our answer
Explanation is provided around line 400.
Tmax, tkeep, and Vcool fabrication parameters control three important factors. Each parameter has an upper and lower limit.
- The sentence in the coclusion "A good superconducting joint was successfully obtained in (100)/(100) joint" is a bit puzzling and not quantitative. In general, the authors could try yo reword the conclusion section.
Our answer
Explanation is provided around line 354.
The difference between “(110)/(110)” and “(100)/(100)” is almost negligible, with a maximum of ±15%

Round 2
Reviewer 3 Report
Comments and Suggestions for Authors
The revisions are proper